

# Depressive symptoms and other risk factors predicting suicide in middle-aged men: a prospective cohort study among Korean Vietnam War veterans

Sang-Wook Yi[1,2] and Jae-Seok Hong[3]

[1] Department of Preventive Medicine and Public Health, Catholic Kwandong University College of Medicine, Gangneung, Republic of Korea
[2] Institute for Clinical and Translational Research, Catholic Kwandong University, Gangneung, Republic of Korea
[3] Department of Healthcare Management, Cheongju University College of Health Sciences, Cheongju, Republic of Korea

Corresponding author
Sang-Wook Yi, flyhigh@cku.ac.kr

## ABSTRACT

**Background.** Few studies have prospectively examined whether depressive symptoms and other risk factors are associated with a higher risk of suicide death in individuals other than high-risk populations such as psychiatric patients and individuals with self-harm histories. The purpose of the study is to prospectively examine whether depressive symptoms assessed by the Beck Depression Inventory (BDI) are associated with greater risk of suicide death and whether depressive symptoms and other risk factors are independent predictors of suicide in general-risk populations. Another aim is to evaluate the sensitivity of the BDI for predicting suicide death.

**Methods.** 10,238 Korean Vietnam War veterans (mean age: 56.3 years) who participated in two surveys in 2001 were followed up for suicide mortality over 7.5 years.

**Results.** 41 men died by suicide. Severely depressed participants had a higher adjusted hazard ratio (aHR = 3.4; 95% CI [1.5–7.7]) of suicide than non-to-moderately depressed ones. Higher suicide risk was associated with more severe depressive symptoms ($p$ for trend = 0.009). After adjustment for depressive symptoms and other factors, very poor health, low education, and past drinking were associated with higher suicide risk, while good health, body mass index, and marital status were not associated with suicide. The sensitivity at the cut-off score of 31 for detecting suicide was higher during the earlier 3.5 years of the follow-up (75%; 95% CI [50–90]) than during the latter 4 years (60%; 95% CI [41–76]).

**Conclusions.** Depressive symptoms are a strong independent predictor and very poor health, low education, and drinking status may be independent predictors of future suicide. The BDI may have acceptable diagnostic properties as a risk assessment tool for identifying people with depression and suicidal potential among middle-aged men.

## INTRODUCTION

Over 800,000 people die by suicide worldwide every year (*World Health Organization, 2014*). Suicide prevention is a high priority in many countries. Understanding the risk factors leading to suicide is crucial for effectively preventing suicide and managing individuals at risk of suicide. Depression has been strongly associated with suicide in retrospective psychological autopsy studies (*Hawton & van Heeringen, 2009*; *Yoshimasu, Kiyohara & Miyashita, 2008*), although an over-underestimation of the association, and biases such as interviewer bias, surrogate interview bias, and recall bias may not be ruled out (*Hjelmeland et al., 2012*). Prospective studies have been conducted mainly among patients with psychiatric disorders, who may not represent the people with psychiatric disorders in the community (*Agerbo, 2007*). Meanwhile, physical health indicators such as health status (*Jee et al., 2011*) and body mass index (BMI) (*Jee et al., 2011*; *Mukamal et al., 2010*), marital status (*Masocco et al., 2010*), drinking status (*Akechi et al., 2006*), and socioeconomic status (*Milner, Page & Lamontagne, 2014*) have also been linked with suicide. Despite many studies supporting these associations, it has seldom been evaluated and is unclear whether each of those factors were independently associated with suicide after adjusting for the others and, notably, for depression, one of the most important known precursors of suicide (*Milner, Page & Lamontagne, 2014*; *Mukamal et al., 2010*).

Meanwhile, suicide risk assessment using depression tools could be relatively easily embedded in the various health care settings to help primary physicians and public health professionals identify people with depression and suicidal potential. However, to the best of our knowledge, the diagnostic characteristics of depression instruments that can be used for suicide risk assessment have never been evaluated in general-risk populations for predicting suicide death.

The primary aim of this study was to prospectively examine whether more severe depressive symptoms assessed by the Korean translation of the Beck Depression Inventory (BDI) would be associated with a greater risk of suicide in middle-aged Korean men, specifically Korean veterans of the Vietnam War. The secondary aim was to evaluate the diagnostic properties of the BDI for predicting suicide death, such as sensitivity, specificity, and area under the receiver operating characteristics (ROC) curve (AUC). The tertiary aim was to examine whether risk factors such as self-rated health status, drinking, BMI, marital status, and socio-economic status were independently associated with suicide after adjusting for each other and for depressive symptoms. Korean Vietnam War veterans have had the same suicide mortality (crude mortality, 35.4 deaths per 100,000 person-years during 1992–2005; standardized mortality ratio = 0.95, 95% CI [0.89–1.02]) expected from the general Korean population in previous research (*Yi & Ohrr, 2011*).

## METHODS

### Study participants

We identified 187,897 Vietnam War veterans during 1999–2000 with the cooperation of the Ministry of Defense and then the Ministry of Government Administration and Home Affairs (*Yi et al., 2014*). Among them, 20,000 veterans were selected by a simple random

sample. After excluding 1,737 individuals who were deceased, had emigrated to another country, or had an unknown residency status as of July 2000, 18,263 were recruited for a health exam or survey. Among them, randomly selected 5,000 veterans (health exam group) were invited to a health exam and a survey by mail on May 4, 2001, and the other 13,263 (survey-only group) received a survey sent out on July 20, 2001. In the end, 10,238 veterans who participated in the health exam or survey (health exam group, $n = 2,005$; survey-only group, $n = 8,233$) were followed up on suicide mortality until December 31, 2008 (Fig. S1). The veterans were provided with a written summary about the research and an invitation to the health exam by post. Return of the postal survey and voluntary participation of health exam were considered implied consent. This study was approved by the Institutional Review Board of Kwandong University (Approval No: KD-13-0105).

## Follow-up and ascertainment of suicide death

The deaths from suicide were ascertained from the 2001–2008 death records of the National Statistical Office. When veterans had emigrated to another country or their residential status was unknown between enrollment and December 2008, the changed date of the residential status was considered the date of a loss to follow-up. A complete follow-up was made for 10,181 veterans (99.4%). Suicide was identified by the International Classification of Diseases, 10th Revision codes X60-X84.

## Measures of depressive symptoms

The self-administered BDI, with 21 items each measured on a 4-point Likert scale (0–3) and a total score range of 0–63, was used to assess depressive symptoms (Beck et al., 1961). The previous Korean-language version of the BDI, translated by two psychiatrists and a clinical psychologist (Rhee et al., 1995), was modified for the present study by two medical doctors, and several graduate students of public health. The internal consistency assessed by Cronbach's alpha was 0.937 in a complete data analysis.

The participants were categorized into two groups (no-to-moderate [0–30] and severe [31–63]) and five groups (depressive symptoms [BDI score]: none [0–13], mild [14–21], moderate [22–30], severe [31–39], and extreme [40–63]) based on quartiles and the highest decile of BDI score. For a sensitivity analysis, they were also categorized based on cut-off score in previous research into two groups (no-to-moderate [0–29] and severe [30–63]) and five groups (none [0–9], mild [10–16], moderate [17–29], severe [30–40], and extreme [41–63]) (Al-Turkait & Ohaeri, 2010; Beck & Steer, 1987; Smarr & Keefer, 2011).

## Measures of other risk factors

Self-rated health, smoking, drinking ("Do you drink alcohol?" [Yes, No. I quit drinking, or Never]), marital status, educational status, and household monthly income were collected through self-reported survey. Self-rated health was assessed by a 5-point Likert scale. Weight and height were measured among participants in health exams, while they were self-reported among the survey-only group. The BMI was obtained from weight divided by the height squared (kg/m$^2$).

## Statistical analysis

Among the 10,238 participants, 3,156 (30.8%) had one or more missing baseline data including smoking ($n = 55$), drinking ($n = 84$), BMI ($n = 102$), self-rated health ($n = 186$), marital status ($n = 143$), educational status ($n = 195$), income status ($n = 426$), or at least one item on the BDI ($n = 2761$). Multiple imputation was performed to estimate the missing baseline data with SAS PROC MI. Markov chain Monte Carlo sampling methods and an expectation–maximization algorithm with a separate chain for each imputation (chain = multiple) were used. 100 imputations were made using all variables analyzed in this study and other variables including military variables (such as Agent Orange exposure index, unit served and military rank during Vietnam service), and self-rated health compared to the same age group. Imputation methods including number of imputations were selected to minimize imputation variance and to ensure the confidence intervals and $p$ values to be reliable (*White, Royston & Wood, 2011*).

The chi-squared test and analysis of variance were performed to compare individual characteristics by two categories of depressive symptoms. A Cox proportional hazards model including various risk factors was implemented to calculate hazard ratios (HRs). The variables included in the Cox model were age at enrollment (years), smoking (current, past, and never smoker), drinking (current, past, and never drinker), BMI (<18.5, 18.5–22.9, 23–24.9, 25–26.9, ≥27), self-rated health (very good or good, fair, poor, and very poor; since no suicide death was observed in those with very good health, those with very good and good health were combined into one group), marital status (living with spouse, living without spouse), education (elementary school or less, middle school, and high school and more), monthly income (less than 1,000,000 Korean won [KRW], 1,000,000–1,990,000 KRW, and 2,000,000 KRW and more, where 1,300 KRW was about one US dollar as of June, 2001) and participant group (survey-only group and health exam group).

The sensitivity (proportion of true positives [suicides] correctly identified) and specificity (proportion of true negatives [non-suicides] correctly identified) at a cut-off score of 31 (or 30) were estimated and their 95% CI was calculated using the Wilson score method (*Newcombe, 1998*). The AUC was estimated using Proc Logistic (ROC statement) and averaged over 100 imputed data. A diagnostic test with an AUC value of 1.0 or 0.5 represents a perfect test or an uninformative test, respectively. Predictive values were not calculated because they are of little use to other populations with different suicide mortality.

The test for trend was done to demonstrate the dose–response relationships of the risk factors (every variable with three or more categories) to suicide by analyzing categories of a risk factor as ordinal variables. Additionally, analyses in all of the participants with follow-up until 2004, and analyses in survivors as of January 1, 2005 with follow-up until 2008, were done to evaluate whether the association of depressive symptoms and other risk factors with suicide, and the diagnostic characteristics of the BDI differ by follow-up period, and these calculations served as sensitivity (robustness) analyses. To test the non-response bias and potential harm of the screening for suicide, the suicide

mortality and age-adjusted hazard ratio (HR) of suicide of the participants ($n = 10,238$) were compared with those of the non-participants ($n = 7,867$).

All statistical analyses were performed using SAS version 9.4 (SAS Inc., Cary, North Carolina, USA). The $p$-value was calculated with two-sided tests. The proportional hazard assumption was tested using Martingale residuals and Schoenfeld residuals in complete data and imputed data analyses. No variables had evidence of a violation of the proportional hazard assumption except for a dummy variable of self-reported health (very good or good, only one suicide) in the complete data analysis.

## RESULTS

### Characteristics of participants

The total follow-up person-years was 73,916. 41 men died by suicide (5.5 per 10,000 person-years, 95% CI [4.1–7.5]) during around 7.5 years of follow-up. The average (SD) age of the Korean veterans was 56.3 (3.5) years at enrollment. Participants with severe depressive symptoms were slightly younger than those with no-to-moderate symptoms, and they tended to be current smokers, past drinkers, slim, living without a spouse, unhealthy, and have little formal education and low earnings, compared to those with no-to-moderate symptoms (Table 1).

### Associations of depressive symptoms and other risk factors with suicide

In the unadjusted analysis, participants with severely depressive symptoms had a higher HR of suicide than those with no-to-moderately depressive symptoms, and increasing HR of suicide was associated with increasing depressive symptoms across the five categories of intensity ($p$ for trend $< 0.001$) (Table 2, Table S1). Additionally, higher suicide risk correlated with a higher BDI score. After adjustment for various risk factors for suicide, the association of depressive symptoms with suicide (although somewhat weaker than before) remained strong.

Unadjusted HRs indicated that very poor health ($p < 0.001$), low education ($p = 0.002$), low household income ($p = 0.03$), past drinking ($p = 0.07$) and living without a spouse ($p = 0.09$) were associated with a higher risk of suicide, while obesity (BMI $\geq 25$) with a lower risk of suicide ($p = 0.07$) (Table 3). In the unadjusted trend test across categories of risk factors, poorer self-rated health, lower education, and lower income were associated with a higher risk of suicide, while more obese subjects were less likely to die by suicide. After adjustment for depressive symptoms and other risk factors, very poor health, low education, and past drinking were still related to a higher HR of suicide, while BMI and marital status were not associated with suicide (Table 3, Table S6). In the adjusted trend test across categories of risk factors, only educational status remained statistically significantly associated with suicide ($p = 0.03$).

The associations of depressive symptoms with suicide were generally stronger during the earlier period of follow-up than during the later period (Table 4, Table S2). The associations of most of the other risk factors with suicide were also generally stronger (although none of them had a $p$-value below 0.05 due to the small number of suicide

**Table 1  Characteristics of the Korean middle-aged male participants by depressive symptoms.**

| Characteristics | | Number (%) | | | |
| --- | --- | --- | --- | --- | --- |
| | | Total | No to moderate symptoms | Severe symptoms | p value[a] |
| | | n = 10,238 | n = 7,470 | n = 2,768 | |
| Age at enrollment | y (SD) | 56.3 (3.5) | 56.5 (3.8) | 56.2 (3.3) | 0.007[b] |
| Smoking status[c] | Never smoker | 3,027 (29.6) | 2,333 (31.2) | 694 (25.1) | <0.001 |
| | Past smoker | 2,480 (24.2) | 1,783 (23.9) | 698 (25.2) | |
| | Current smoker | 4,730 (46.2) | 3,354 (44.9) | 1,377 (49.7) | |
| Drinking status | Never drinker | 1,216 (11.9) | 782 (10.5) | 434 (15.7) | |
| | Past drinker | 2,675 (26.1) | 1,756 (23.5) | 919 (33.2) | <0.001 |
| | Current drinker | 6,347 (62.0) | 4,932 (66.0) | 1,415 (51.1) | |
| BMI (kg/m$^2$) | Below 18.5 | 323 (3.2) | 160 (2.1) | 163 (5.9) | |
| | 18.5–22.9 | 4,118 (40.2) | 2,843 (38.1) | 1,275 (46.0) | <0.001 |
| | 23.0–24.9 | 2,992 (29.2) | 2,296 (30.7) | 696 (25.2) | |
| | 25 or above | 2,805 (27.4) | 2,171 (29.1) | 634 (22.9) | |
| Marital status[c] | Living with spouse | 9,377 (91.6) | 7,060 (94.5) | 2,317 (83.7) | <0.001 |
| | Living without spouse | 861 (8.4) | 410 (5.5) | 450 (16.3) | |
| Self-rated health[c] | Very good or good | 734 (7.2) | 724 (9.7) | 11 (0.4) | |
| | Fair | 3,748 (36.6) | 3,405 (45.6) | 343 (12.4) | <0.001 |
| | Poor | 4,503 (44.0) | 2,944 (39.4) | 1,559 (56.3) | |
| | Very poor | 1,253 (12.2) | 398 (5.3) | 856 (30.9) | |
| Educational status[c] | Elementary school or below | 2,407 (23.5) | 1,489 (19.9) | 918 (33.2) | |
| | Middle school | 2,782 (27.2) | 1,934 (25.9) | 848 (30.6) | <0.001 |
| | High school or above | 5,049 (49.3) | 4,048 (54.2) | 1,002 (36.2) | |
| Household income per month[c] (Korean Won)[d] | Below 1,000,000 | 3,450 (33.7) | 1,852 (24.8) | 1,597 (57.7) | |
| | 1,000,000–1,990,000 | 4,693 (45.8) | 3,707 (49.6) | 987 (35.6) | <0.001 |
| | 2,000,000 or more | 2,095 (20.5) | 1,911 (25.6) | 184 (6.7) | |
| Participant group | Survey-only group | 8,233 (80.4) | 5,881 (78.7) | 2,352 (85.0) | <0.001 |
| | Health exam group | 2,005 (19.6) | 1,589 (21.3) | 416 (15.0) | |
| Depressive symptoms (1)[e] (BDI score) | No (0–13) | 2,360 (23.0) | 2,360 (31.6) | 0 (0.0) | |
| | Mild (14–21) | 2,554 (24.9) | 2,554 (34.2) | 0 (0.0) | |
| | Moderate (22–30) | 2,556 (25.0) | 2,556 (34.2) | 0 (0.0) | <0.001 |
| | Severe (31–39) | 1,627 (15.9) | 0 (0.0) | 1,627 (58.8) | |
| | Extreme (40–63) | 1,141 (11.1) | 0 (0.0) | 1,141 (41.2) | |
| Depressive symptoms (2)[f] (BDI score) | No (0–9) | 1,307 (12.8) | 1,307 (17.5) | 0 (0.0) | |
| | Mild (10–16) | 2,625 (25.6) | 2,625 (35.1) | 0 (0.0) | |
| | Moderate (17–29) | 3,281 (32.0) | 3,281 (43.9) | 0 (0.0) | <0.001 |
| | Severe (30–40) | 2,015 (19.7) | 257 (3.4) | 1,758 (63.5) | |
| | Extreme (41–63) | 1,010 (9.9) | 0 (0.0) | 1,010 (36.5) | |

**Notes.**

BDI, Beck Depression Inventory; BMI, body mass index; SD, standard deviation.

[a] Combined p-value of chi-squared analysis over multiple imputed data.

[b] Combined p-value of analysis of variance over multiple imputed data.

[c] Sum of the number of participants from both groups may not equal the total due to rounding of averages over multiple imputed data.

[d] 1,300 Korean Won was about one US dollar as of June, 2001.

[e] Cut-off score based on quartiles and the last decile.

[f] Cut-off score based on previous research.

**Table 2  Hazard ratios of suicides by depressive symptoms in Korean middle-aged men (n = 10,238).[a]**

| Categories of depressive symptoms (score) | | No. of suicides[c] | Unadjusted analysis | | Adjusted analysis[b] | |
|---|---|---|---|---|---|---|
| | | | *p*-value | HR (95% CI) | *p*-value | HR (95% CI) |
| Total BDI score (0–63) | Five-score increase | 41 | **<0.001** | 1.4 (1.2–1.5) | **0.01** | 1.2 (1.1-1.5) |
| 2-group analysis[d] | No-to-moderate (0–30) | 14 | | 1.0 (Reference) | | 1.0 (Reference) |
| | Severe (31–63) | 27 | **<0.001** | 5.4 (2.8–10.6) | **0.004** | 3.4 (1.5–7.7) |
| 5-group analysis[d] | No (0–13) | 3 | | 1.0 (Reference) | | 1.0 (Reference) |
| | Mild (14–21) | 6 | 0.51 | 1.6 (0.4–6.8) | 0.69 | 1.4 (0.3–6.1) |
| | Moderate (22–30) | 5 | 0.70 | 1.4 (0.3–6.3) | 0.94 | 0.9 (0.2–5.0) |
| | Severe (31–39) | 12 | **0.009** | 5.4 (1.5–19.4) | 0.12 | 3.3 (0.7–14.4) |
| | Extreme (40–63) | 15 | **<0.001** | 10.2 (3.0–35.0) | 0.05 | 4.6 (1.0–21.5) |
| | Trend test[e] | 41 | **<0.001** | 1.9 (1.5–2.5) | **0.009** | 1.6 (1.1–2.2) |

**Notes.**

BDI, Beck Depression Inventory; CI, confidence interval; HR, hazard ratio.

[a] Hazard ratios were calculated using a Cox proportional hazards model over multiple imputed data.

[b] Adjusted for age at enrollment, smoking status, drinking status, body mass index, self-rated health, marital status, educational status, household monthly income and participant group.

[c] Number of suicides may not match those of the other groups of depressive symptoms due to rounding of averages over multiple imputed data.

[d] Cut-off score based on quartiles and the last decile.

[e] Five categories of Depressive symptoms were analyzed as ordinal variables.

deaths) during the earlier period than the later period, while educational status was more strongly associated with suicide during the later period than the earlier period (Table S3).

### Diagnostic properties of the Beck Depression Inventory

For the unbinned total score and cut-off score of 31 or above, respectively, the AUC values were 0.71 and 0.69 during the follow-up, and they were higher during the first half of follow-up (2001–2004) than during the latter half (2005–2008) (Table 5, Fig. S2 and Table S4). The overall sensitivity and specificity of the 2-group test at a cut-off score of 31 or above on predicting suicide during 7.5 years of follow-up were 66% (95% CI [50–78]) and 73% (95% CI [72–74]), respectively. The sensitivity decreased over the follow-up period, while the specificity remained relatively unchanged. The AUC analyses showed that cut-off score of 31 or above had the highest AUC values across cut-off scores of BDI for predicting suicide (Table S5).

## DISCUSSION

### Depressive symptoms and suicide

Severe depressive symptoms were more strongly associated with suicide compared to no-to-moderate depressive symptoms in this study, after adjusting for other potential risk factors for suicide, although the association was weaker than that of depressive disorders examined in psychological autopsy studies (*Yoshimasu, Kiyohara & Miyashita, 2008*). The association of depressive symptoms with suicide was stronger during the first 3.5 years of follow-up than during the latter 4-year period in accordance with previous research (*Bramness et al., 2010*; *Sun et al., 2012*). This indicates that a cohort study of long follow-up with one baseline assessment and without considering the change in depressive symptoms

**Table 3** Hazard ratios of suicides by risk factors in Korean middle-aged men ($n = 10{,}238$).[a]

| Categories of risk factors | | No. of suicides[c] | Unadjusted analysis | | Adjusted analysis[b] | |
|---|---|---|---|---|---|---|
| | | | *p*-value | HR (95% CI) | *p*-value | HR (95% CI) |
| Age at enrollment | One year increase in age | 41 | 0.24 | 0.9 (0.9–1.0) | 0.27 | 0.9 (0.8–1.0) |
| Smoking status | Never smoker | 9 | | 1.0 (Reference) | | 1.0 (Reference) |
| | Past smoker | 7 | 0.95 | 1.0 (0.4–2.6) | 0.51 | 0.7 (0.3–2.0) |
| | Current smoker | 25 | 0.15 | 1.8 (0.8–3.8) | 0.47 | 1.3 (0.6–2.9) |
| | *p* for trend[d] | 41 | 0.11 | 1.4 (0.9–2.0) | 0.36 | 1.2 (0.8–1.8) |
| Drinking status | Never drinker | 1 | | 1.0 (Reference) | | 1.0 (Reference) |
| | Past drinker | 16 | 0.07 | 8.4 (0.8–86.9) | 0.05 | 8.7 (1.0–75.5) |
| | Current drinker | 23 | 0.19 | 4.8 (0.5–49.1) | 0.10 | 6.1 (0.7-53.0) |
| | *p* for trend[d] | 41 | 0.80 | 1.1 (0.7–1.7) | 0.29 | 1.3 (0.8–2.1) |
| Body mass index (kg/m$^2$) | Below 18.5 | 2 | 0.79 | 1.2 (0.3–5.2) | 0.75 | 0.8 (0.2-3.4) |
| | 18.5-22.9 | 22 | | 1.0 (Reference) | | 1.0 (Reference) |
| | 23.0–24.9 | 10 | 0.20 | 0.6 (0.3–1.3) | 0.55 | 0.8 (0.4–1.7) |
| | 25 or above | 7 | 0.07 | 0.5 (0.2–1.1) | 0.23 | 0.6 (0.2–1.4) |
| | *p* for trend[d] | 41 | **0.04** | 0.7 (0.5–1.0) | 0.29 | 0.8 (0.6–1.2) |
| Self-rated health | Very good or good | 1 | 0.35 | 0.4 (0.1–2.9) | 0.92 | 1.1 (0.1–10.3) |
| | Fair | 9 | 0.33 | 0.7 (0.3–1.5) | 0.71 | 1.2 (0.5–2.9) |
| | Poor | 16 | | 1.0 (Reference) | | 1.0 (Reference) |
| | Very poor | 15 | **<0.001** | 3.6 (1.8–7.3) | **0.02** | 2.4 (1.1–5.2) |
| | *p* for trend[d] | 41 | **<0.001** | 2.4 (1.6–3.6) | 0.13 | 1.5 (0.9–2.5) |
| Marital status | Living with spouse | 35 | | 1.0 (Reference) | | 1.0 (Reference) |
| | Living without spouse | 6 | 0.09 | 2.1 (0.9–5.0) | 0.76 | 1.2 (0.5–2.8) |
| Educational status | Elementary school or below | 16 | **0.002** | 3.6 (1.6–8.2) | **0.04** | 2.4 (1.0–5.6) |
| | Middle school | 15 | **0.02** | 2.8 (1.2–6.3) | 0.09 | 2.1 (0.9–4.9) |
| | High school or above | 10 | | 1.0 (Reference) | | 1.0 (Reference) |
| | *p* for trend[d] | 41 | **<0.001** | 0.6 (0.4–0.8) | **0.03** | 0.7 (0.5–1.0) |
| Household income per month (Korean Won)[e] | Below 1,000,000 | 21 | **0.03** | 3.7 (1.1–12.2) | 0.76 | 1.2 (0.3–4.7) |
| | 1,000,000–1,990,000 | 16 | 0.29 | 2.0 (0.6–6.8) | 0.79 | 1.2 (0.3–4.4) |
| | 2,000,000 or more | 4 | | 1.0 (Reference) | | 1.0 (Reference) |
| | *p* for trend[d] | 41 | **0.01** | 0.5 (0.3–0.9) | 0.80 | 0.9 (0.5–1.7) |
| Participant group | Health exam group | 7 | 0.46 | 0.7 (0.3–1.7) | 0.77 | 0.9 (0.4–2.1) |
| | Survey-only group | 34 | | 1.0 (Reference) | | 1.0 (Reference) |

**Notes.**

BDI, Beck Depression Inventory; CI, confidence interval; HR, hazard ratio.

[a] Hazard ratios were calculated using a Cox proportional hazards model over multiple imputed data.

[b] Variables included in the Cox model were age at enrollment, smoking status, drinking status, body mass index, self-rated health, marital status, educational status, household monthly income, participant group, and depressive symptoms (5 categories based on quartiles and the last decile of the total BDI score).

[c] Sum of the number of suicides may not equal the total suicides ($n = 41$) due to rounding of averages over multiple imputed data.

[d] Categories of risk factors were analyzed as ordinal variables.

[e] 1,300 Korean Won was about one US dollar as of June, 2001.

during the follow-up, may underestimate the effects of depressive symptoms on suicide (*Bramness et al., 2010*). Additionally, in the current study, higher suicide risk was associated with a higher depressive symptom score in accordance with previous cohort studies in general-risk populations (*Bramness et al., 2010*; *Sun et al., 2012*). Overall, this prospective cohort study confirmed that depressive symptoms were a strong predictor of future suicide.

**Table 4** Adjusted hazard ratios of suicides by depressive symptoms according to follow-up period.[a]

| Categories of depressive symptoms (score) | | From 2001 to 2004 ($n = 10{,}238$) | | | From 2005 to 2008 ($n = 9{,}877$) | | |
|---|---|---|---|---|---|---|---|
| | | No. of suicides[b] | p-value | HR (95% CI) | No. of suicides[b] | p-value | HR (95% CI) |
| Total BDI score (0–63) | Five-score increase | 16 | **0.02** | 1.3 (1.1–1.8) | 25 | 0.16 | 1.1 (1.0–1.4) |
| 2-group analysis[c] | No-to-moderate (0–30) | 4 | | 1.0 (Reference) | 10 | | 1.0 (Reference) |
| | Severe (31–63) | 12 | **0.02** | 5.2 (1.3–19.8) | 15 | 0.06 | 2.7 (0.9–7.6) |
| 5-group analysis[c] | No (0–13) | 1 | | 1.0 (Reference) | 2 | | 1.0 (Reference) |
| | Mild (14–21) | 3 | 0.70 | 1.6 (0.1–17.6) | 3 | 0.87 | 1.2 (0.2–8.4) |
| | Moderate (22–30) | 1 | 0.99 | 0.0 | 4 | 0.80 | 1.3 (0.2–9.2) |
| | Severe (31–39) | 4 | 0.33 | 3.3 (0.3–37.8) | 8 | 0.21 | 3.3 (0.5–20.9) |
| | Extreme (40–63) | 8 | 0.13 | 7.1 (0.6–88.3) | 7 | 0.23 | 3.3 (0.5–23.8) |
| | Trend test[d] | 16 | **0.03** | 1.9 (1.1–3.3) | 25 | 0.09 | 1.4 (0.9–2.2) |

Notes.

BDI, Beck Depression Inventory; CI, confidence interval; HR, hazard ratio.

[a] Hazard ratios were calculated using a Cox proportional hazards model over multiple imputed data, after adjustment for age at enrollment, smoking status, drinking status, body mass index, selfrated health, marital status, educational status, household monthly income and participant group.

[b] Number of suicides may not match those of the other groups of depressive symptoms due to rounding of averages over multiple imputed data.

[c] Cut-off score based on quartiles and the last decile.

[d] Five categories of depressive symptoms were analyzed as ordinal variables.

**Table 5** Diagnostic characteristics of Beck Depression Inventory by follow-up period in Korean middle-aged men.

| Follow-up period | Characteristics | Cut-off of 31 or above[a] Rate (95% CI)[b] | Unbinned total score Rate (95% CI)[b] |
|---|---|---|---|
| From 2001 to 2008 | Sensitivity[c] | 66 (50–78) | |
| | Specificity[d] | 73 (72–74) | |
| | AUC[e] | 0.69 (0.62–0.77) | 0.71 (0.63–0.80) |
| From 2001 to 2004 | Sensitivity[c] | 75 (50–90) | |
| | Specificity[d] | 73 (72–74) | |
| | AUC[e] | 0.74 (0.63–0.85) | 0.75 (0.61–0.90) |
| From 2005 to 2008 | Sensitivity[c] | 60 (41–76) | |
| | Specificity[d] | 74 (73–75) | |
| | AUC[e] | 0.67 (0.57–0.77) | 0.69 (0.58–0.80) |

Notes.

AUC, area under the receiver operating characteristics curve; CI, confidence interval.

[a] Cutoff score based on third quartile.

[b] 95% CI was calculated using Wilson score method.

[c] Proportion of true positives (suicides) correctly identified.

[d] Proportion of true negatives (non-suicides) correctly identified.

[e] A diagnostic test with an AUC value of 1.0 or 0.5 represents a perfect test or an uninformative test, respectively.

## Diagnostic characteristics of Beck Depression Inventory

To the best of our knowledge, this is the first study that has reported diagnostic characteristics for suicide death in a general-risk population. Although studies on suicidal ideation or attempt have been performed, attempters may share some properties with

suicides; however, they may be two different populations, especially among males (*Isometsa & Lonnqvist, 1998*; *Parra Uribe et al., 2013*). The sensitivity and specificity at the cut-off score of 31 or above of the BDI in Korean middle-aged men for predicting future suicide, especially during the earlier period of follow-up, was comparable at least to those of other tools such as the Beck Hopelessness Scale (sensitivity 80%, specificity 42%) (*McMillan et al., 2007*), Beck Suicide Intent Scale (sensitivity 77%, specificity 49%) (*Harriss & Hawton, 2005*), and a tool constructed from stepwise discriminant analysis of various instruments (sensitivity 56%, specificity 74%) (*Pokorny, 1983*) that were mainly applied to inpatients with psychiatric disorders or those who had done self-harm. They are also comparable to those of BDI at cut-off score of 23 or above in psychiatric outpatients (sensitivity 76%, specificity 62%) (*Beck et al., 1990*) and those of BDI-II at cut-off score of 31 or above in prisoners (for attempted suicide or self-harm incident; sensitivity 80%, specificity 69%) (*Perry & Gilbody, 2009*). Although there is a room for improvement, the BDI may have an acceptable sensitivity and specificity as a suicide risk assessment tool.

It has been reported that male patients rarely initiated suicide-related discussions with their physician (*Vannoy & Robins, 2011*), while physicians were also reluctant to discuss suicide for various reasons including the fear of stimulating suicide in patients with depression by asking about it (*Crawford et al., 2011*; *Schulberg et al., 2004*). The suicide mortality of the participants in the present study (41 suicides, 5.5 suicides per 10,000 person-years) was not higher than that of the non-participants (34 suicides in 7,867 men, 6.0 (95% CI [4.3–8.4]) suicides per 10,000 person-years), and the age-adjusted hazard ratio of suicide (0.93, 95% CI [0.59–1.46]) also was not higher in the participants compared to non-participants. These results suggest that asking people about depressive symptoms using the BDI does not seem to be associated with a higher risk of suicide.

## Other risk factors and suicide

Although physical health, marital status, drinking, and socioeconomic status have been associated with suicide in previous research, their association with suicide has rarely been examined after adjustment for depression (*Conwell, Duberstein & Caine, 2002*), especially in a prospective manner. Marital status was not associated with suicide in clinical patients with depression in a systematic review (*Hawton et al., 2013*), and it was also not associated with suicide after adjusting for other risk factors in a large psychological autopsy study in China (*Phillips et al., 2002*). Although BMI was inversely associated with suicide in large cohort studies, depressive symptoms were not accounted for in those studies (*Jee et al., 2011*; *Mukamal et al., 2010*). A review also concluded that much, if not all, of the association of physical health factors with suicide was mediated by affective disorders (*Conwell, Duberstein & Caine, 2002*).

In prospective studies, evidence linking regular alcohol consumption with suicide death is lacking (*Jee et al., 2011*; *Mukamal et al., 2007*), except for serious alcohol-related conditions such as alcohol use disorder and very heavy drinking (*Akechi et al., 2006*; *LeardMann et al., 2013*). Meanwhile, higher risks of suicide in past drinkers have also been identified in Japanese middle-aged men (depression was not adjusted for, though, in the Japanese

study) (*Akechi et al., 2006*). As for socioeconomic status, despite a strong dose–response relationship of both educational status and household income with suicide in unadjusted analysis, the association of household income was substantially weakened after adjustment for self-rated health and depressive symptoms, because household income status was strongly associated with self-rated health and depressive symptoms in the current study.

Our findings suggest that very poor health, low education, and drinking status may be independently associated with suicide, while good or fair health, BMI, household income, and marital status may not be factors independently related factors to suicide. Instead, the associations of these non-independent factors with suicide may rather be mediated through depressive symptoms and other factors in Korean middle-aged men.

### Limitations of the study

Despite of the strengths of the study, such as nearly complete prospective follow-up on suicide using national mortality data and the analysis of the diagnostic characteristics of a potential suicide risk assessment tool on suicide death in a general-risk population, this study has some limitations. First, our findings are based on 41 suicides, so the study may have lacked statistical power in some analyses. In particular, risk factors having a weak association with suicide may not be clearly identified in this study. Second, in cohort studies, nonresponse to the baseline survey may introduce bias. However, the suicide mortality of 7,867 (43%) non-respondents did not differ from that of participants, and item non-response to questionnaires was minimized with multiple imputation of 100 datasets. Third, the validity of the suicide death listed on death certificates was not examined separately. Since any misclassification on the suicide death could be most likely non-differential according to depressive symptoms, potential misclassifications would not substantially overestimate the hazard ratios.

It may be a limitation of its generalizability that this study's participants were Korean Vietnam War veterans. However, among US military personnel, mental health problems but not military-specific variables were independent risk factors for suicide (*LeardMann et al., 2013*). Additionally, the suicide mortality in Korean Vietnam War veterans did not differ from that from the male population of Korea (*Yi & Ohrr, 2011*). Therefore, we believe that performing the study with Vietnam War veterans did not substantially hinder its generalizability. Meanwhile, although psychological disorders such as depression have been universal risk factors among many cultural groups (*Sun et al., 2012*; *Zhang et al., 2004*), the strength of association and the association in itself of some risk factors with suicide may differ by ethnicity, culture, age, and gender (*Bjerkeset, Romundstad & Gunnell, 2008*; *Goldston et al., 2008*; *Milner et al., 2013*; *Sun et al., 2012*; *Zhang et al., 2004*). Therefore, some results in Korean middle-aged men may not be generalizable to other ethnic, cultural, or age groups, or to females.

### CONCLUSIONS

Although the association was weaker than estimated from psychological autopsy studies, this study prospectively confirmed that depressive symptoms are a strong independent predictor of suicide. The sensitivity and specificity at the BDI cut-off score of 31 or above in

Korean middle-aged men for predicting future suicide during 2001–2008 were 66% (95% CI [50–78]) and 73% (95% CI [72–74]), respectively. They were 75% (95% CI [50–90]) and 73% (95% CI [72–74]), respectively, for predicting future suicide during 2001–2004. BDI may have an acceptable sensitivity and specificity as a suicide risk assessment tool to help physicians and public health professionals identify individuals with depression and suicidal potential. This study also suggests that very poor self-rated health, drinking status and low education, but not marital status and BMI, may be independent predictors of suicide.

**Abbreviations**

| | |
|---|---|
| **AUC** | area under the receiver operating characteristics curve |
| **BDI** | Beck Depression Inventory |
| **BMI** | body mass index |
| **CI** | confidence interval |
| **HR** | hazard ratio |
| **KRW** | Korean Won |
| **ROC** | the receiver operating characteristics |

## ACKNOWLEDGEMENT

The authors truly thank the staff of the National Statistical Office of Korea for providing the national mortality data.

### Funding

This work was supported by a research grant of the Ministry of Patriots and Veterans Affairs (MPVA) of Korea. MPVA provided administrative support in the data collection. Except for a research grant and administrative support in the data collection, MPVA had no role in study design, in the analysis and interpretation of data, in the writing of the report, or in the decision to submit the article for publication.

### Grant Disclosures

The following grant information was disclosed by the authors:
Ministry of Patriots and Veterans Affairs (MPVA).

### Competing Interests

The authors declare there are no competing interests.

### Author Contributions

- Sang-Wook Yi conceived and designed the experiments, performed the experiments, analyzed the data, contributed reagents/materials/analysis tools, wrote the paper, prepared figures and/or tables, reviewed drafts of the paper.
- Jae-Seok Hong performed the experiments, reviewed drafts of the paper.

## Human Ethics

The following information was supplied relating to ethical approvals (i.e., approving body and any reference numbers):

The veterans were provided with a written summary about the research and an invitation to the health exam by post. Return of the postal survey and voluntary participation of health exam were considered implied consent. This study was approved by the Institutional Review Board of Kwandong University (Approval No: KD-13-0105).

## Supplemental Information

Supplemental information for this article can be found online at http://dx.doi.org/10.7717/peerj.1071#supplemental-information.

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
