# Peer review of "Depressive symptoms and other risk factors predicting suicide in middle-aged men: a prospective cohort study among Korean Vietnam War veterans"

_PeerJ, doi:10.7717/peerj.1071_

## Round 0.1 · original submission · Major Revisions

Thank you for considering PeerJ for the publication of your work. As you can see from the appended reviewer comments a number of sharp criticsms have been made during the review process which need to be addressed. We think your paper is interesting and could be reconsidered after substantial revision and hope you will be able to submit a revised manuscript addressing the criticisms that have been made.

·

Basic reporting

Title: Suggest to mention in the title that the population studied were Korean veterans of the Vietnam War, rather than simply middle-aged Korean men, as this would make the article clearly of more interest to a more specific audience.

Abstract: clearly articulated and succinct

Introduction: well written and summarizes what is already known

Methods: recommend a review by a statistician

Results and Tables and Figures: clearly labeled and easy to interpret. I recommend a review by a statistician

Conclusion: see below comments under 'Validity of the findings'

Experimental design

1. Longitudinal observational cohort study with large sample size and high rates of participation. However, the self reported measures (BDI) were only measured at one time point (baseline).

2. Selected military-based population - high risk subjects for mental health problems.

3. Subject inclusion and recruitment seem appropriate for this type of study

Validity of the findings

Depressive disorders are nowadays considered chronic conditions which run a remitting and relapsing or sometimes persistently chronic course, and the biggest risk of suffering from current depressive symptoms, is a past history of having had a depressive disorder. Suicidal ideation is one of the symptoms of severe depression, so it is not surprising that there is a strong association between severity of depressive symptoms in the past and subsequent suicide.

One of the merits of this study was the large study sample size, who were followed longitudinally for 7.5 years, to examine the naturalistic outcomes for depressive symptoms.

The authors claim that the BDI can be used as a suicide screening tool needs to be better justified. I am not sure you can claim that the BDI a screening instrument for suicide based on an association between a positive BDI score and subsequent death from suicide which may have occurred years later. EG It is possible that if the BDI had been administered in the month before the suicide occurred, that the BDI score may have been negative. I think it is fairer to say that there is a strong association between a positive BDI and subsequent suicide, and at best the BDI can be used for future risk stratification.

Additional comments

Suggest to change the terminology from increased/ decreased risk of... to higher/ lower association with...
Suggest to change the BDI as a screening tool, to a future risk stratification tool

Reviewer 2 ·

Basic reporting

- This article has two purposes. One is to examine the relation between suicide death and depressive symptom (BDI) or other risk factors, and the other is to evaluate the sensitivity of BDI tool. But I am doubtful that these two purposes can be achieved at once within an article. Moreover, analysis for the former purpose has been carried out without evaluation of the tool (latter purpose). Beause of this reversal or dual-purpose, more than three categories of BDI score group are used in this article: 5-group based on quantiles, 5-group based on previous research, 2-group based on quantiles, and continuous or ordinal variables.

- The title of this article says "in Korean middle-aged men" and abstract says "in general population". But the stduy population of this article is Vietnam War veterans during 1999-2000.Though the authors propose the previous research on the difference of suicide mortality, it would be so scanty as to satisfy the premise.

Experimental design

- BDI tool is the fundamental material of this article. Among 10,238 cases, 2,761 cases (27.0%) have one item or more missing responds, and the authors have performed multiple imputation to estimate the missing data. The rationale for choosing that imputation method should be explained.

- Definitions of some variables are uncertain. The range of the drinking status categorization ('never drinker', 'past drinker', 'current drinker') and the rationale of the asymmetrical 4-group categrization of SRH Likert scale should be explained.

- Weight and height were measured among some participants and self-reported among other particpants. I am doubtful that two different measuring method can be combined for use, and that this discrepancy can be rectified by creating 'participant group' variable.

- Among 10,238 participant, only 41 men died by suicide (0.4%). Considering this low incidence, some variables seems to be excessively divided. For example, the number of suicide of 'Never drinker' group is 1, that of 'below 18.5 BMI' group is 2, and that of 'very good or good' SRH group is 1.

Validity of the findings

- Despite of the strength as a prospective cohort study, the finding that "Depressive symptoms are a predictor of suicide" has much widely been accepted as to be an high orginality.

- The findings on the relation between other risk factors and suicides are to be modified with revision of experimental design. And some findings are to be discussed in depth: Why past drinking - not current drinking - elevate suicide risk? Why 'poor' group is referenced in SRH HR analysis? Why the adjusted economic status (household income) is not significant compared with previous research (Or is the categrization of income level adequate)?

- In regard to sensitivity evaluation, "cut-off of 31" (3rd quantile) is the optimal point? In other words, what is the optimal cut-off point from the unbinned (continuous) total score AUC analysis?

Additional comments

"No Comments"

---

## Round 0.2 · accepted · Accept

The additional information is valuable to the reader and addresses all my concerns. I look forward to seeing this interesting report in publication.